The effect of icotinib or apatinib on the pharmacokinetic profile of oxycodone in rats and the underlying mechanism

Zhou Qi 1
Ye Feng 1
Ye Zhize 2
Gao Nanyong 1
Kong Qihui 1
Hu Xiaoqin 1
Qian Jianchang 1 qianjc@wmu.edu.cn
Wu Bin 3 drwzyywb@163.com
1 Wenzhou Medical University , Wenzhou , China
2 Shaoxing People’s Hospital , Shaoxing , China
3 The First Affiliated Hospital of Wenzhou Medical University , Wenzhou , China
Santillo Michael
Electronic publication date: 2023 Dec 8
Publication date: 2023
Volume: 11
Electronic Location ID: e16601
Received 2023 Aug 16; Accepted 2023 Nov 15
Copyright: © 2023 Zhou et al.
Copyright year: 2023
Copyright holder: Zhou et al.
License: This is an open access article distributed under the terms of the Creative Commons Attribution License, which permits unrestricted use, distribution, reproduction and adaptation in any medium and for any purpose provided that it is properly attributed. For attribution, the original author(s), title, publication source (PeerJ) and either DOI or URL of the article must be cited.
License URL: https://creativecommons.org/licenses/by/4.0/

Keywords: Icotinib, Apatinib, Oxycodone, UPLC-MS/MS, Interaction

Funding: National Key Research and Development Program of China 2020YFC2008301 National Natural Science Foundation of China 81973397 Natural Science Foundation of Zhejiang Province LTGC23H310001 Wenzhou Municipal Science and Technology Bureau Y20220192 This work was supported by the National Key Research and Development Program of China (2020YFC2008301), the National Natural Science Foundation of China (81973397), the Natural Science Foundation of Zhejiang Province (LTGC23H310001) and the project of the Wenzhou Municipal Science and Technology Bureau (Y20220192). The funders had no role in study design, data collection and analysis, decision to publish, or preparation of the manuscript.

==============================
This study aimed to investigate the interactions between icotinib/apatinib and oxycodone in rats and to unveil the underlying mechanism. An ultra-performance liquid chromatography–tandem mass spectrometry (UPLC-MS/MS) method was developed and validated to determine oxycodone and its demethylated metabolite simultaneously. In vivo, Sprague–Dawley (SD) male rats were administered oxycodone with or without icotinib or apatinib. Blood samples were collected and subjected to UPLC-MS/MS analysis. An enzyme incubation assay was performed to investigate the mechanism of drug–drug interaction using both rat and human liver microsomes (RLM and HLM). The results showed that icotinib markedly increased the AUC(0–t) and AUC(0–∞) of oxycodone but decreased the CLz/F. The Cmax of oxycodone increased significantly upon co-administration of apatinib. In vitro, the Km value of oxycodone metabolism was 101.7 ± 5.40 μM and 529.6 ± 19.60 μM in RLMs and HLMs, respectively. Icotinib and apatinib inhibited the disposition of oxycodone, with a mixed mechanism in RLM (IC50 = 3.29 ± 0.090 μM and 0.95 ± 0.88 μM, respectively) and a competitive and mixed mechanism in HLM (IC50 = 22.34 ± 0.81 μM and 0.48 ± 0.05 μM, respectively). In conclusion, both icotinib and apatinib inhibit the metabolism of oxycodone in vitro and in vivo. Therefore, the dose of oxycodone should be reconsidered when co-administered with icotinib or apatinib.

Introduction

Comorbidities and multiple medicines are common during cancer therapy. Therefore, the incidence of drug–drug interactions is high in clinical practice (Ahn et al., 2017; Akbulut & Urun, 2020; Yeung, Gubili & Mao, 2018). However, compared with the purpose of prolonging the life of terminally ill patients, the adverse reactions that occur during drug therapy are often overlooked. This situation undoubtedly reduces the patients’ quality of life (Fink & Gallagher, 2019; Neufeld, Elnahal & Alvarez, 2017). Thus, it is important to study the drug–drug interactions in cancer treatment.

Cancer pain is one of the most common comorbidities of cancer. Its incidence is 39.3% after curative treatment, 55% during anticancer treatment, and 66% in advanced or metastatic status (Das et al., 2009; van den Beuken-van Everdingen et al., 2016). Oxycodone, a semisynthetic opioid, has been used as a potent analgesic for more than a century (Kalso, 2005; Kinnunen et al., 2019). CYP3A4 and CYP2D6 are the primary enzymes involved in the metabolism of oxycodone and produce its main metabolite desmethyloxycodone (Cai et al., 2021; Kummer et al., 2011; Söderberg Löfdal, Andersson & Gustafsson, 2013). Therefore, drugs that affect the activities of CYP3A4 and CYP2D6 would lead to the stratification of oxycodone blood exposure.

Tyrosine kinase receptors are widely used in clinical therapeutics (Ahn et al., 2017; Yoneda et al., 2019). Among them, icotinib and apatinib are two representative drugs (Du et al., 2021; Meng et al., 2020). Icotinib is the first oral inhibitor of epidermal growth factor receptor (EGFR), and it is prescribed for EGFR-positive non-small cell lung cancer (NSCLC) (Guan, He & Li, 2014; Tan et al., 2023). Common adverse reactions of icotinib include rash, diarrhea, and pruritus (Shi et al., 2015). Apatinib is another representative drug used in the treatment of NSCLC and advanced gastric cancer (Xue et al., 2018; Yang et al., 2022). There is in vitro evidence that icotinib inhibits CYP3A (Chen et al., 2015; Zhuang et al., 2016). Furthermore, apatinib can inhibit the activity of many cytochrome P450 enzymes, including CYP2D6 (Ye et al., 2022; Zhou et al., 2014). Therefore, the co-administration of icotinib or apatinib with oxycodone could result in a fluctuation of drug response.

In this study, we employed a library of tyrosine kinase inhibitors (TKIs) to conduct a comprehensive screening for drug–drug interactions. We successfully identified icotinib and apatinib as highly potent inhibitors, based on their significant inhibitory level on oxycodone metabolism. Furthermore, we evaluated the drug–drug interactions through in vivo experiments. Additionally, we utilized rat liver microsomes (RLMs) and human liver microsomes (HLMs) to investigate the pharmacokinetic inhibition of oxycodone by icotinib and apatinib. The outcomes of this investigation could provide valuable theoretical support for precision medicine, enabling the avoidance of adverse reactions and ineffective treatments.

Materials and Methods

Chemicals and reagents

We obtained standard substances noroxycodone and midazolam (internal standard, IS) from Sigma-Aldrich (St. Louis, MO, USA). Additionally, we purchased icotinib from Inviwo Chemical Technology (Guangzhou, China) and acquired apatinib from Sunflower Technology Development Co., Ltd. (Beijing, China). Liquid chromatography (LC)–grade acetonitrile (ACN) was obtained from Merck (Darmstadt, Germany). Ultrapure water was produced using a Milli-Q purification system. Finally, we obtained HLMs and RLMs from Corning Life Sciences Co., Ltd. (Beijing, China).

UPLC-MS/MS conditions

An ultra-performance liquid chromatography–tandem mass spectrometry (UPLC-MS/MS) method was developed to detect oxycodone and noroxycodone as previously described (Lin et al., 2022). This method utilized a Waters XEVO TQD triple quadrupole mass spectrometer with an electrospray ionization source (Milford, MA, USA). Sample separation was achieved using a BEH C18 column, and incubation was carried out at 40 °C. The mobile phase consisted of 0.1% formic acid (A) and acetonitrile (B), with a gradient elution at a flow rate of 0.40 mL/min for 3.0 min. The gradient conditions were as follows: 10%–90% B (0–1.0 min), 90%–10% B (1.0–2.0 min), and 10% B (2.1–3.0 min). Multiple reaction monitoring was performed in the positive ion mode.

Microsome incubation assay

A 200-μL culture system was used, which included 1 M phosphate buffer, 0.2 mg/mL RLM or HLM, 1 mM NADPH, and 8–2,000 μM oxycodone. Before measurement, the mixture without NADPH was pre-incubated at 37 °C for 5 min. Then, 1 mM NADPH was added, and the reaction was terminated by cooling to −80 °C after 30 min. Next, 400 μL acetonitrile and 20 μL midazolam (200 ng/mL, IS) were added to the mixture. After shaking for 2 min, the supernatant was obtained by centrifugation at 13,000 rpm for 10 min and subjected to UPLC-MS/MS analysis. With this method, we obtained the Km values of oxycodone in RLMs and HLMs.

To evaluate the inhibitory effects of 24 TKIs on oxycodone metabolism, the concentration of each drug was set at 100 μM and added to the abovementioned culture system. Moreover, based on the Km values, the concentration of oxycodone was also set at 100 μM. With this method, we obtained the inhibition rates of these 24 drugs on oxycodone metabolism and identified icotinib and apatinib as those with high inhibition rates. To determine the half-maximal inhibitory concentration (IC50) of icotinib and apatinib, the concentrations of icotinib and apatinib were set at 0.01, 0.1, 1, 10, 25, 50, and 100 μM, while the concentration of oxycodone was set at 100 μM (in RLMs) and 500 μM (in HLMs), based on the corresponding Km values. To determine the inhibition mode, the concentration of oxycodone was set at 25, 50, 100, and 200 μM in RLMs and at 50, 250, 500, and 1,000 μM in HLMs. The concentration of icotinib was set at 0, 0.75, 1.5, and 3 μM in RLMs and at 0, 5, 10, and 20 μM in HLMs, based on the IC50 value. The concentration of apatinib was set at 0, 0.25, 0.5, and 1 μM in RLMs and at 0, 0.125, 0.25, and 0.5 μM in HLMs. After incubation, the samples were prepared and measured by UPLC-MS/MS in accordance with the abovementioned method. Data were collected as previously described (Cai et al., 2021).

Animals and ethical statement

Male Sprague–Dawley (SD) rats were obtained from Viton Lever Laboratory Animal Center (Beijing, China). Rats with a body weight of 200 ± 20 g were used for the experiment. The rats were acclimated in animal rooms for 1 week, with the room temperature maintained at 20–25 °C and the humidity ranged from 50% to 65%. To simulate day and night cycles, the light and dark conditions were alternated every 12 h. During the acclimation period, the rats were provided with unlimited access to food and water. The study was conducted in accordance with the guidelines of the Ethics Committee of Wenzhou Medical University (wydw2020-0322). The experimental animal center has a license number SCXK (Zhejiang, China) 2019-0001. Prior to the start of the experiment, the rats were fasted for 12 h. For euthanasia, 2% isoflurane was used, while 5% isoflurane was used for pharmacokinetic experiments.

Animal experiments

We randomly divided 18 SD rats into three groups (n = 6), namely the control group, the icotinib + oxycodone group, and the apatinib + oxycodone group. We dissolved icotinib and apatinib in 0.5% CMC-Na solution to a final concentration of 37.5 and 40 mg/mL, respectively. Based on the bioequivalent dose, we converted the concentrations of the two inhibitors. The icotinib + oxycodone group received 37.5 mg/kg icotinib per mouse by oral administration, while the apatinib + oxycodone group received 40 mg/kg apatinib per mouse by gastric gavage. After 30 min, 3 mg/kg oxycodone was subcutaneously injected into each mouse in the three groups. To dilute oxycodone to a final concentration of 2 mg/mL before injection, we used saline solution. At 0.083, 0.167, 0.25, 0.5, 1, 2, 4, 6, 8, and 10 h after injection, we collected blood samples from the tail vein as previously described (Ye et al., 2023). Each sample was centrifuged at 13,000 rpm for 10 min, and 100 μL of the supernatant was mixed with 20 μL midazolam (200 ng/mL) and 200 μL acetonitrile. After shaking for 2 min, the samples were centrifuged again at 13,000 rpm for 10 min. Finally, we collected the supernatant for UPLC-MS/MS analysis.

Statistical analysis

The kinetic curve was analyzed using GraphPad Prism 5.0 software and fitted using the Lineweaver–Burk double reciprocal plot method to compare the logarithm of the inhibitor with the normalized response. The pharmacokinetic parameters were determined using Drug and Statistics (DAS) software 3.0, utilizing a noncompartmental model. The drug–time curves were plotted using Origin 8.0. All data are presented as the mean ± standard deviation. Statistical analysis was performed using an independent-samples t test, with the significance level set at P < 0.05.

Results

Development of UPLC-MS/MS assay to determine the concentrations of oxycodone and noroxycodone

The monitoring transitions of oxycodone, noroxycodone, and midazolam are m/z 316.2 → 241.1, m/z 302.2 → 187, and m/z 326.1 → 291.1, respectively. The chromatogram condition was optimized. As shown in Fig. 1, there was no obvious endogenous interference. The retention time for oxycodone, noroxycodone, and IS was 1.06, 1.01, and 1.23 min, respectively.

Figure 1 Representative chromatograms of oxycodone, noroxycodone and IS.

(A) The blank plasma sample; (B) the blank plasma sample spiked with oxycodone, noroxycodone and IS; (C) rat plasma sample after administration.

Clarifying the drug interaction profile of oxycodone

Next, we explored drugs that could potentially interact with oxycodone. In the RLM incubation system, the Km value of oxycodone was 101.7 ± 5.40 μM, as shown in Fig. 2A. In the HLM incubation system, the Km value of oxycodone was 529.6 ± 19.60 μM, as shown in Fig. 2B. To determine the interaction between TKIs and oxycodone, 24 TKIs were selected and incubated with oxycodone. As shown in Fig. 2C, apatinib and icotinib had the highest efficacy in suppressing oxycodone metabolism, with an inhibitory rate of 99.33% and 97.11%, respectively. Therefore, we selected these two drugs as inhibitors to further study their inhibitory effect on oxycodone metabolism. We determined the IC50 values of icotinib/apatinib in RLMs and HLMs (Tables 1 and 2, Fig. 3). The results showed that oxycodone metabolism was concentration-dependently inhibited by apatinib in RLMs, with an IC50 of 0.95 ± 0.88 μM. Icotinib also inhibited the disposition of oxycodone, with IC50 = 3.29 ± 0.09 μM. In HLMs, the IC50 values for apatinib and icotinib were 0.48 ± 0.05 μM and 22.34 ± 0.81 μM, respectively.

Figure 2 Michaelis constant (Km) in RLM (rat liver microsomes)/HLM (human liver microsomes) and comparison of the inhibitory effects of drugs on the metabolism of oxycodone in RLM.

(A) Km in RLM. (B) Km in HLM. The incubition assay was performed as indicated as in the method. (C) Comparison of the inhibitory effects of drugs on the metabolism of oxycodone in RLM. Data are presented as the means ± SD, n = 3.

Table 1 The IC50 values and inhibitory effects of apatinib and icotinib on oxycodone in RLMs.

RLM	IC50 (μM)	Inhibition type	Ki (μM)	αKi (μM)	α	
Apatinib	0.95 ± 0.88	Mixed inhibition	0.65	1.81	2.71	
Icotinib	3.29 ± 0.09	Mixed inhibition	1.04	9.07	8.73	

Table 2 The IC50 values and inhibitory effects of apatinib and icotinib on oxycodone metabolism in HLMs.

HLM	IC50 values (μM)	Inhibition type	Ki (μM)	αKi (μM)	α	
Apatinib	0.48 ± 0.05	Mixed inhibition	1.04	0.41	0.26	
Icotinib	22.34 ± 0.81	Competitive inhibition	9.42			

Figure 3 Half-maximal inhibitory concentration (IC50) value in RLM/HLM.

(A) Apatinib with various concentrations for IC50 in the activity of RLM. (B) Apatinib with various concentrations for IC50 in the activity of HLM. (C) Icotinib with various concentrations for IC50 in the activity of RLM. (D) Icotinib with various concentrations for IC50 in the activity of HLM. Data are presented as the means ± SD, n = 3.

Effects of icotinib and apatinib on oxycodone metabolism in rats

To further evaluate the interaction between icotinib/apatinib and oxycodone in vivo, the rats were administered oxycodone with or without icotinib or apatinib. As shown in Fig. 4 and Table 3, the AUC(0–t) and AUC(0–∞) values of oxycodone in the oxycodone + icotinib group were significantly increased by 1.4 and 1.2 times, respectively, compared with those in the control group (P < 0.05). In addition, the CLz/F value decreased by about 0.7 times (P < 0.05). Upon co-administration of apatinib and oxycodone, the Cmax value of oxycodone significantly increased (by 1.5 times) compared with the control group (P < 0.05).

Figure 4 The concentration-time curve of oxycodone in the experimental and control groups.

(A) Icotinib and (B) apatinib. Data are presented as the means ± SD, n = 6.

Table 3 Pharmacokinetic parameters of oxycodone in three groups.

		Control	oxycodone + apatinib	oxycodone + icotinib	
AUC(0-t)	ug/L*h	452.10 ± 88.11	521.20 ± 114.10	647.00 ± 167.50*	
AUC(0-∞)	ug/L*h	467.30 ± 83.00	543.40 ± 110.30	658.60 ± 156.10*	
MRT(0-t)	h	1.87 ± 0.19	1.69 ± 0.23	2.08 ± 0.33	
MRT(0-∞)	h	2.31 ± 0.46	2.29 ± 0.43	2.38 ± 0.39	
t1/2z	h	2.66 ± 0.55	3.32 ± 1.29	2.04 ± 1.38	
Tmax	h	0.67 ± 0.26	0.58 ± 0.20	0.83 ± 0.26	
Vz/F	L/kg	25.91 ± 9.22	27.80 ± 12.93	15.79 ± 14.25	
CLz/F	L/h/kg	6.60 ± 1.22	5.69 ± 1.04	4.77 ± 1.12*	
Cmax	ug/L	231.80 ± 56.22	350.20 ± 86.25*	239.00 ± 33.98	
Notes:

Compared to control group.

* P < 0.05.

Icotinib and apatinib inhibit the metabolism of oxycodone through a mixed mechanism

As shown in Fig. 5 and Tables 1 and 2, both apatinib and icotinib inhibited oxycodone metabolism in RLMs through a mixed mechanism. The Ki value was 0.65 μM for apatinib and 1.04 μM for icotinib. In HLMs, the inhibition mode of apatinib was the same as that in RLMs, with Ki = 1.04 μM, whereas icotinib competitively inhibited the metabolism of oxycodone, with Ki = 9.42 μM.

Figure 5 Lineweaver-Burk plot and the secondary plot for Ki in RLM/HLM.

Lineweaver-Burk plot and the secondary plot for Ki in the inhibition of oxycodone. (A) Icotinib and (B) apatinib with various concentrations in RLMs. Lineweaver-Burk plot and the secondary plot for Ki in the inhibition of oxycodone metabolism by (C) icotinib and (D) apatinib with various concentrations in HLMs. Data are presented as the means ± SD, n = 3.

Discussion

Patients with cancer often take multiple drugs, which increases the chances for drug interactions (Kummer et al., 2011). Oxycodone is widely used in cancer patients, but its adverse effects are often overlooked compared with prolongation of patients’ life. It has been demonstrated that oxycodone is mainly metabolized by CYP3A4 and CYP2D6 (Werk & Cascorbi, 2014; Zhou, 2008). Therefore, the plasma concentration and tissue distribution of oxycodone may be affected by drugs that inhibit CYP3A4 and/or CYP2D6. Indeed, it has been shown that many drugs (e.g., voriconazole and diphenhydramine) can interact with oxycodone (Hagelberg et al., 2009; Sadiq et al., 2011).

Currently, TKIs are the most commonly used targeted therapy for cancer (Huang, Jiang & Shi, 2020). Based on this, there is a possibility of combining apatinib or icotinib with oxycodone in cancer patients. Previous laboratory experiments have shown that icotinib has an inhibitory effect on CYP3A (Zhang et al., 2018). Moreover, it has been reported that apatinib inhibits the activity of cytochrome P450 enzymes (Bao et al., 2018; Zhang et al., 2020). To further investigate this aspect, we conducted in vitro experiments using RLMs and HLMs. We found that both apatinib and icotinib strongly inhibited the effects of oxycodone in both types of microsomes.

To investigate the interaction between these two inhibitors and oxycodone, we conducted experiments in SD rats. The results showed that co-administration of icotinib and oxycodone significantly increased the AUC(0-t) and AUC(0-∞) values of oxycodone and decreased the CLz/F value by approximately 0.7-fold. Additionally, apatinib significantly increased the Cmax value of oxycodone. These findings suggest that icotinib and apatinib enhance the exposure to oxycodone in rats, and icotinib has a significant impact on oxycodone clearance in vivo. This indicates that icotinib and apatinib likely exert inhibitory effects, possibly through their strong inhibition of liver enzyme CYP3A4, thereby leading to decreased enzyme activity and inhibition of oxycodone metabolism. Previous literature reports have shown that the toxic blood concentration of oxycodone is 0.06 mg/L (Darke, Duflou & Torok, 2011), while after oral administration of 40 mg oxycodone in fasting individuals, the toxic blood concentration is reported to be 40.2 ± 10.8 ng/mL (Ito et al., 2022), which represents a difference of approximately 1.5 times. In our experiment, compared with the control group, the AUC values of the combined apatinib and icotinib groups differed by 1.2 and 1.4 times, respectively. Therefore, AUC analysis suggests that although the two TKIs do meet the criteria for moderate CYP inhibitors, they may still affect the toxicity of oxycodone.

To further investigate the inhibitory effects of these two inhibitors on oxycodone, we examined their inhibitory mechanisms in RLMs and HLMs. As for the experimental design, adding ice-cold acetonitrile directly to the sample can effectively terminate the reaction. However, in experiments with large samples, this procedure may not be feasible in a timely manner. We conducted comparisons and found that the results obtained with the freezing method had a smoother linearity compared to adding ice-cold acetonitrile directly. The findings revealed that in RLMs, both icotinib and apatinib inhibited the metabolism of oxycodone through a mixed mechanism. In HLMs, icotinib acted as a competitive inhibitor, while apatinib acted as a mixed inhibitor. Interestingly, there was inconsistency between their inhibitory abilities in vivo and in vitro, particularly for apatinib. According to previous literature, the reported Cmax value of apatinib is 382.30 ± 46.70 ng/mL (equivalent to 0.96 ± 0.12 μM) (Lou et al., 2016), which differed slightly from the Ki value in RLMs (0.65 μM). This discrepancy may explain why apatinib exhibited strong inhibitory effects in vitro but a weaker inhibitory ability in vivo.

In summary, this study examined the interaction between two TKIs (icotinib and apatinib) and oxycodone. The results showed that both TKIs inhibited the metabolism of oxycodone in RLMs and HLMs, leading to the changes in its pharmacokinetic characteristics. Thus, cancer patients may experience adverse reactions of varying degrees when oxycodone is used alongside icotinib or apatinib. In clinical practice, it is important to carefully consider and monitor the concurrent use of these drugs, as the pharmacokinetics of oxycodone could be significantly altered even at therapeutic doses. Therefore, the dosage of oxycodone should be appropriately reduced. However, it is important to note that this study focused primarily on the impact of genetic polymorphisms and drug interactions on oxycodone metabolism and did not take gender into account as an additional variable. Hence, the findings of this experiment can only be applied to male individuals. Overall, our study provides accurate guidance for the combined use of oxycodone and TKIs such as icotinib or apatinib.

Supplemental Information

Supplemental Information 1 Experimental raw data.

Click here for additional data file.

We thank the Scientific Research Center of Wenzhou Medical University for the consultation and instrument availability that supported this work. We thank LetPub for its linguistic assistance during the preparation of this manuscript.

Additional Information and Declarations

Competing Interests

Author Contributions

Animal Ethics

Data Availability

The authors declare that they have no competing interests.

Qi Zhou conceived and designed the experiments, performed the experiments, analyzed the data, prepared figures and/or tables, and approved the final draft.

Feng Ye conceived and designed the experiments, performed the experiments, analyzed the data, authored or reviewed drafts of the article, and approved the final draft.

Zhize Ye conceived and designed the experiments, performed the experiments, authored or reviewed drafts of the article, and approved the final draft.

Nanyong Gao conceived and designed the experiments, performed the experiments, prepared figures and/or tables, and approved the final draft.

Qihui Kong conceived and designed the experiments, analyzed the data, prepared figures and/or tables, and approved the final draft.

Xiaoqin Hu conceived and designed the experiments, analyzed the data, prepared figures and/or tables, and approved the final draft.

Jianchang Qian conceived and designed the experiments, analyzed the data, authored or reviewed drafts of the article, and approved the final draft.

Bin Wu conceived and designed the experiments, analyzed the data, authored or reviewed drafts of the article, and approved the final draft.

The following information was supplied relating to ethical approvals (i.e., approving body and any reference numbers):

Ethics Committee of Wenzhou Medical University.

The following information was supplied regarding data availability:

The raw measurements are available in the Supplemental Files.

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
