# Peer review of "The effect of icotinib or apatinib on the pharmacokinetic profile of oxycodone in rats and the underlying mechanism"

_PeerJ, doi:10.7717/peerj.16601_

## Round 0.1 · original submission · Major Revisions

For your invited resubmission following a previous rejection, three new reviewers have commented on your manuscript and suggest major revisions.

The discussion section, in particular, needs to be significantly improved. Currently, the discussion only repeats the results and does not provide any interpretations or meaning for the study. The discussion should mention the limitations of using rat models, potential clinical significance, etc. Given the results, it's not clear if they indicate potential interactions in humans.

Reviewer 1 ·

Basic reporting

Authors were performed in vitro and in vivo experiments to clarify the drug-drug interaction between oxycodone and TKIs.
I think this manuscript can become better. I have a few suggestions to improve it.
1. A space before might be needed at the start of parentheses in some parts of the manuscript (e.g., line # 83, 86, 89, 91, 93, 96, 97, 99, 100, 102, 104, 105 and so on.)
2. I think the name of the city should be capitalized on line 156 (i.e., Zhejiang).
3. I agree that SD rats are a good model system to evaluate the PK of some compounds, but it is also clear that there are many species differences. I am wondering if the results can "undoubtedly provide theoretical support". I think the results from animal experiments are a good clue for the mechanism, but it could be doubted by other researchers. Even in this study, the Km values in rats and humans were different for oxycodone (lines 195, and 196).
4. UPLC would stand for "ultra-performance liquid chromatography", not a generic "liquid chromatography" on line 125.
5. Line 208: I think the description is not valid. I think "given oxycodone with or without icotinib and apatinib" would be relevant instead of "given icotinib and apatinib with or without oxycodone".
6. According to Figure 2, anlotinib showed a higher inhibitory effect than icotinib. How icotinib was selected instead of anlotinib? Also, the terminology 'potency' is different from 'efficacy'. The experiment described on line 141 and 142 might be a way to measure 'efficacy', even though the authors wrote 'apatinib and icotinib were the most potent' on line 198.

Experimental design

7. Line 136 to 137: The method for quenching the microsomal assay seemed not adequate. According to the manuscript, the reaction was terminated by cooling to the deep freezer (-80 Celsius degree), and then 400 uL acetonitrile was added to the mixture. The temperature of the mixture would not be dropped instantly to -80 degrees, which means the measured concentration might not be the proxy of the true concentration at 30 minutes. In my understanding, many researchers quench their reaction by spiking the reaction mixture into a cold organic solvent (e.g., acetonitrile). Please check your steps again.
8. Line 202: I think the authors have seen 'concentration-dependent' not 'dose-dependent'. This paragraph is not about in vivo study.

Validity of the findings

9. In terms of PK parameter change, it would be better to suggest statistical significance (e.g., t-test between two groups).
10. Also, I am wondering if a 0.7-fold change is critical for oxycodone toxicity. For example, US FDA categorizes moderate CYP inhibitor which molecule can increase AUC from 2- to 5-fold (Clinical Drug Interaction Studies — Cytochrome P450 Enzyme- and Transporter-Mediated Drug Interactions Guidance for Industry).
11. The first figure on the panel A of Figure 1, there was a peak at 1.23 minute. I am wondering if the peak height of midazolam is significantly higher than the endogenous peak at 1.23 minutes.

Reviewer 2 ·

Basic reporting

The manuscript by Zhou et al describes the effect of two tyrosine kinase inhibitors (TKI) on oxycodone pharmacokinetics, demonstrating some small differences between with and without these TKI.

Unfortunately the manuscript was very poorly written in English and requires extensive editing by someone proficient in English. See below for some points:

Line 41: “This study aimed to investigate the interactions between icotinib/apatinib and oxycodone in rats, and to unveil the mechanism underlied.” This should read: …the underlying mechanism
Line 48: “Increased obviously” is not a quantifiable amount. Please correct.
Line 54: “Therefore, the dose of oxycodone should be reconsidered when co-administrated with icotinib and apatinib.” Should be co-administered, not co-administrated.
Line 84: “medicine companied adverse reactions” doesn’t make sense.
Line 109: TKI abbreviation not described previously. Please correct.
Line 170: please clarify if ‘gavage’ means ‘oral gavage’ for route of administration and why that route of administration was chosen.
Line 234 states that bioequivalent doses of TKIs were used, but no references were used. This information would be more helpful in the methods section.
Line 244 repeats the Line 234 statement on bioequivalency


Literature references were sparse.

The background/context was OK.

Tables/Figures were OK.

The results suggest that oxycodone levels may need to be adjusted given the effect of TKI on small changes. However - it is not clear to this reviewer how relevant these effects would be in humans.

Experimental design

No comment.

Validity of the findings

The discussion section is very sparse and mostly repeats the results section. For instance:
"The results of present study demonstrated that icotinib increases the AUC(0-t) and AUC(0- >) of oxycodone. Furthermore, the CLz/F is decreased by about 0.7 times. In addition, we found apatinib increase the Cmax of the oxycodone."

This information is a result, not a finding/discussion.

And places where the results seemed to indicate something interesting, such as below, is hard to interpret - as the authors do not interpret these results:
"The pharmacokinetics parameters showed that the Cmax of apatinib is 382.30 ± 46.70 ng/mL (0.96 ± 0.12 ¿M), which is close to IC50 and Ki obtained in RLM."

Most of the discussion section is spent repeating the background, repeating the results, and making some concluding statements.

Additional comments

The authors would do well to have the manuscript edited by someone proficient in English, as well as re-writing the discussion section, focusing on interpretation of the results rather than re-stating the results. Comparing their current findings to information that is already in the literature would help as well.

·

Basic reporting

In the article icotinib or apatinib was used in conjunction with oxycodone respectively. So it is suggested the title "icotinib or apatinib" instead of "icotinib and apatinib" affects the metabolism of oxycodone drugs.

Experimental design

1.Please elaborate the reason why oxycodone was subcutaneous injected into rats of groups B and C after 30 minutes of gavage, such as half-life period,and so on.
2.When icotinib or apatini is used in combination with oxycodone that primarily metabolize by CYP3A4 or CYP2C9 in liver, the pharmacokinetics is changed. It is suggested to test how the icotinib or apatinib effected on CYP3A activity in rats.

Validity of the findings

The article didn‘t document the effect of icotinib or apatinib on pharmacokinetics of oxycodone. It is suggested to illustrate whether to increase or decrease the dosage of oxycodone when combining two drugs in clinic, and what is the specific guiding significance for clinical practice

Additional comments

None.

---

## Round 0.2 · Minor Revisions

The reviewers recommend a few more minor revisions to be made, which will improve understanding. Also, please address Reviewer #1, points 7 and 10 more carefully. Both of these points need to be clearly and explicitly noted in the discussion section and signposted so that the field can make their own judgements.

Reviewer 2 ·

Basic reporting

The article is much improved. However, although the authors claimed that the manuscript was reviewed by a fluent English speaker, there are still many many grammatical and English-based errors. While these errors may not reduce the impact of the report itself (i.e., one can at least understand what the authors mean and are saying), it reduces the quality of the report. I have noted a few instances below with their wording in quotations and my revisions below. These are only a handful of the noted instances. The authors may need to pay an editorial service to properly edit these. Again, I will consider this as a 'minor' form of needed improvement because it is readable as is. All other issues were amended.

“This study aimed to investigate the interactions between icotinib/apatinib and oxycodone in rats, and to unveil the mechanism underlied.”

This study aimed to investigate the interactions between icotinib/apatinib and oxycodone in rats, and to unveil the underlying mechanism.

“The results showed that icotinib markedly increased the AUC(0-t) and AUC(0-∞) of oxycodone, oppositely CLz/F decreased” needs a period.

“There had evident that icotinib takes a certain inhibitory effect on CYP3A in vitro”

There is evidence that icotinib has an inhibitory effect on CYP3A in vitro

“Furthermore, literatures showed that apatinib can inhibit the activity of many cytochrome P450 enzymes including CYP2D6”

Furthermore, literature shows that apatinib can inhibit the activity of many cytochrome P450 enzymes including CYP2D6

“As shown in Figure 2C, apatinib and icotinib had the highest efficacy in suppressing the oxycodone metabolizing with the inhibitory rate of 99.33% and 97.11%.”

As shown in Figure 2C, apatinib and icotinib had the highest efficacy in suppressing oxycodone metabolism with inhibitory rates of 99.33% and 97.11%.

“Furthermore, as shown in Figure 5, Table 1 and 2, both apatinib and icotinib inhibit oxycodone metabolism underlied mixed mechanism in the RLM.”

Furthermore, as shown in Figure 5, Table 1 and 2, both apatinib and icotinib inhibit oxycodone metabolism underlie the mixed mechanism in the RLM.

Experimental design

No comment

Validity of the findings

No comment

·

Basic reporting

The article has been revised according to the requirements, and it is recommended to further polish the language, such as using the passive voice in the “Chemicals and reagents”.

Experimental design

no comment

Validity of the findings

no comment

---

## Round 0.3 · accepted · Accept

The authors improved the English language through a professional editing service, and also addressed comments in the discussion.